# Perceptions of distinctions between patient and healthcare zones among intensive care unit nurses at a Korean tertiary hospital: A cross-sectional study

Hayoung Chang[1], JaHyun Kang[1,2]*

1 College of Nursing, Seoul National University, Seoul, Republic of Korea, 2 Research Institute of Nursing Science, Seoul National University, Seoul, Republic of Korea

* jahyunkang@gmail.com

**Data Availability Statement:** All relevant data are within the manuscript and its Supporting Information files. - Our submission contains all raw data required to replicate the results of our study.

## Abstract

### Background

Intensive care unit (ICU) patients face higher infection risks from invasive procedures, highlighting the critical role of ICU nurses in infection prevention. Clear differentiation between the patient and healthcare zones is essential for effective hand hygiene and disinfection, yet research on this topic is limited.

### Aim

To assess ICU nurses' perception of the concept of patient and healthcare zones and evaluate their similarity and accuracy in identifying the zones.

### Methods

A descriptive survey was conducted at a 2,732-bed tertiary hospital in Korea from 28 July to 27 August 2022. Participants were recruited from various ICUs through flyers. 225 questionnaires—with illustrations 27 item locations for three ICU scenarios—were made available at nursing stations for voluntary completion. Participants were asked to classify items into the patient zone or the healthcare zone. Similarity scores reflected participant agreement, while accuracy scores measured the proportion of correct answers. Participants' free-text opinions regarding zone classification were analysed thematically.

### Findings

104 nurses participated voluntarily. Average similarity and accuracy were 84.7% and 82.7%, respectively. The top 8 items, with over 97% similarity and accuracy, were all frequently in contact with ICU patients (e.g., pulse oximeter, Levin tube, central line, urine bag, and patient bed). The bottom 7 items, with less than 80%, included the glucometer, flashlight, trolley, and sink. Participants with higher education levels had significantly higher similarity ($p = .044$) and accuracy ($p = .033$), whereas those already familiar with the patient-zone concept had significantly higher accuracy ($p = .009$). From the free-text analysis,

Please check the uploaded raw data file (under "other" item).

**Funding:** The author(s) received no specific funding for this work.

**Competing interests:** The authors have declared that no competing interests exist.

participants considered factors beyond proximity to the patient, such as patient contact, room type, and distance.

## Conclusions

To address zone classification challenges, clear criteria for patient and healthcare zones, space redesign, and educational programs are recommended. Further research is necessary to improve greater clarity and consensus regarding patient and healthcare zones to enhance infection control practices.

## Introduction

The European Centre for Disease Prevention and Control (ECDC) estimated 4.5 million healthcare-associated infection (HAI) episodes from 2016 to 2017 in acute care hospitals in the European Union (EU) and European Economic Area (EEA) [1]. Indeed, healthcare environments have been underscored as potential reservoirs for pathogens that significantly transmit HAIs [2–16]. Colonised or infected patients can shed pathogens into their surroundings, and these microorganisms can survive for extended periods on items and surfaces [3].

To prevent pathogen transmission and reduce HAIs, the World Health Organization introduced the 'Five Moments for Hand Hygiene' model, which has been adopted globally for training, monitoring, and reporting among healthcare personnel (HCP) across various healthcare settings [17, 18]. This model incorporates the concept of two distinct zones within the healthcare environment: patient and healthcare [17]. The patient zone revolves around patient X and their immediate surroundings, encompassing the patient's intact skin and all inanimate surfaces directly or indirectly contacted by the patient [17]. To prevent the transmission of HAI pathogens, all items entering and leaving the patient zone should be decontaminated [17, 18]. Conversely, the healthcare zone includes all surfaces outside the patient's designated area [17].

Nevertheless, the complex nature of ongoing tasks, resource constraints, and the diversity of settings have generated confusion among HCP regarding the proper indication of hand hygiene timing and the clear differentiation of zones [19–24]. In healthcare settings, a lack of congruent perception regarding patient zones, which clarifies the indications for hand hygiene, may result in unintentional pathogen transmission [25].

Intensive care unit (ICU) patients are at a higher risk of developing HAIs owing to their severe conditions and exposure to invasive devices [26]. In the EU and EEA, the ECDC estimated the prevalence of patients with at least one HAI in ICUs as 19.2%, the highest number compared with all other specialties combined (5.2%) [1]. As nurses typically have the most extensive direct contact with patients in hospitals [27], ICU nurses play a crucial role in infection control. A study involving 200 ICU nurses at a tertiary hospital in Korea reported that 72% of participants identified 'after touching a patient's surroundings' as being the most ambiguous moment for hand hygiene [22]. Among these respondents, 41% attributed their ambiguity to the challenge of distinguishing the patient zone from the healthcare zone [22]. Another study conducted in a general ward in Switzerland reported limited accuracy (68%) and similarity (77%) among doctors and nurses who were tasked with classifying 32 item cards into either patient or healthcare zones [24].

Given the limited prior research on this topic and the importance of HCP's zone perceptions for infection control, we aimed to examine how the zone concept is applied in daily care activities by ICU nurses and obtain insights for improving infection control practices like

hand hygiene or item disinfection. The study objectives were to assess ICU nurses' perceptions of the zone concept and evaluate their accuracy and similarity in identifying the zones by presenting scenarios and illustrations depicting various patient care situations with item use.

## Methods

This descriptive survey was conducted at a 2,732-bed tertiary hospital in Korea, from 28 July to 27 August 2022. Ethics approval was obtained from the Asan Medical Center Institutional Review Board. Nurses with at least 2 months of independent work experience providing direct patient care within ICUs were eligible for this study. Nurse administrators who did not engage in direct patient care were also excluded. Participants were recruited from various ICUs—medical, surgical, neurological, neurosurgical, and paediatric—through flyers posted at nursing stations. A minimum of 180 participants was needed to analyse significant differences based on general characteristics. To account for a 20% dropout rate, the target sample size was set at 225 participants. Accordingly, 225 printed consent forms and questionnaires were made available at the ICU nursing stations, allowing nurses to access and complete them voluntarily. Written informed consent was collected from all participants.

The study questionnaire developed for this study comprised basic demographics, a main survey, and additional opinions (see S1 File, the English version of the questionnaire). Definitions of patient and healthcare zones were provided at the beginning of the questionnaire. First, the demographic questions included age, gender, work experience, department, education level, prior knowledge of patient zone concepts, and learning route. Second, for the main survey, 47 items associated with the risk of infection transmission were initially included in the literature review [8–16]. The researcher with ICU work experience created three scenarios depicting the use of these items: (1) assessing a patient's condition and measuring vital signs; (2) patient bathing and measuring body weight; (3) measuring blood glucose levels and administering medication. These scenarios were chosen because they include routine care activities that ICU nurses commonly perform using the selected items. While the patient in Scenario 1 was situated in an open ICU area, the patient in Scenarios 2 and 3, who was the same individual, was placed in a private room. To prevent confusion among respondents regarding nursing care scenarios and item locations, the researcher created illustrations for three scenarios providing the locations of the 47 items. Content validity was assessed by five area experts, all of whom held a master's degree or higher: one nursing professor with infection prevention and control (IPC) expertise, two IPC practitioners, and two ICU nurses with more than 5 years of experience. Items were rated as follows: 4 points for highly relevant, 3 points for somewhat relevant, 2 points for minimally relevant, and 1 point for not relevant. After eliminating 17 items with a content validity index of 0.75 or less and 3 items with similar characteristics, the final 27 items were included (Fig 1). Throughout the main survey, participants were tasked with categorising each item into the patient zone, healthcare zone, or 'difficult to determine' option with a brief description of their reason. Third, seven open-ended questions were posted to elicit the participants' logic in decision-making. Following each scenario, three identical questions were presented, prompting participants to consider whether, if the situation changed, any items in the corresponding scenario might belong to the opposite zone compared to their initial selection and to explain the reason for the change. At the end of the survey, four additional open-ended questions were asked: (1) participants' most significant criteria when categorising items into the patient or healthcare zone; (2) challenges encountered when differentiating between the two zones during regular work; (3) suggestions for addressing the identified challenges; (4) specific items that were difficult to categorise and confusing when determining the appropriate time for disinfection or hand hygiene.

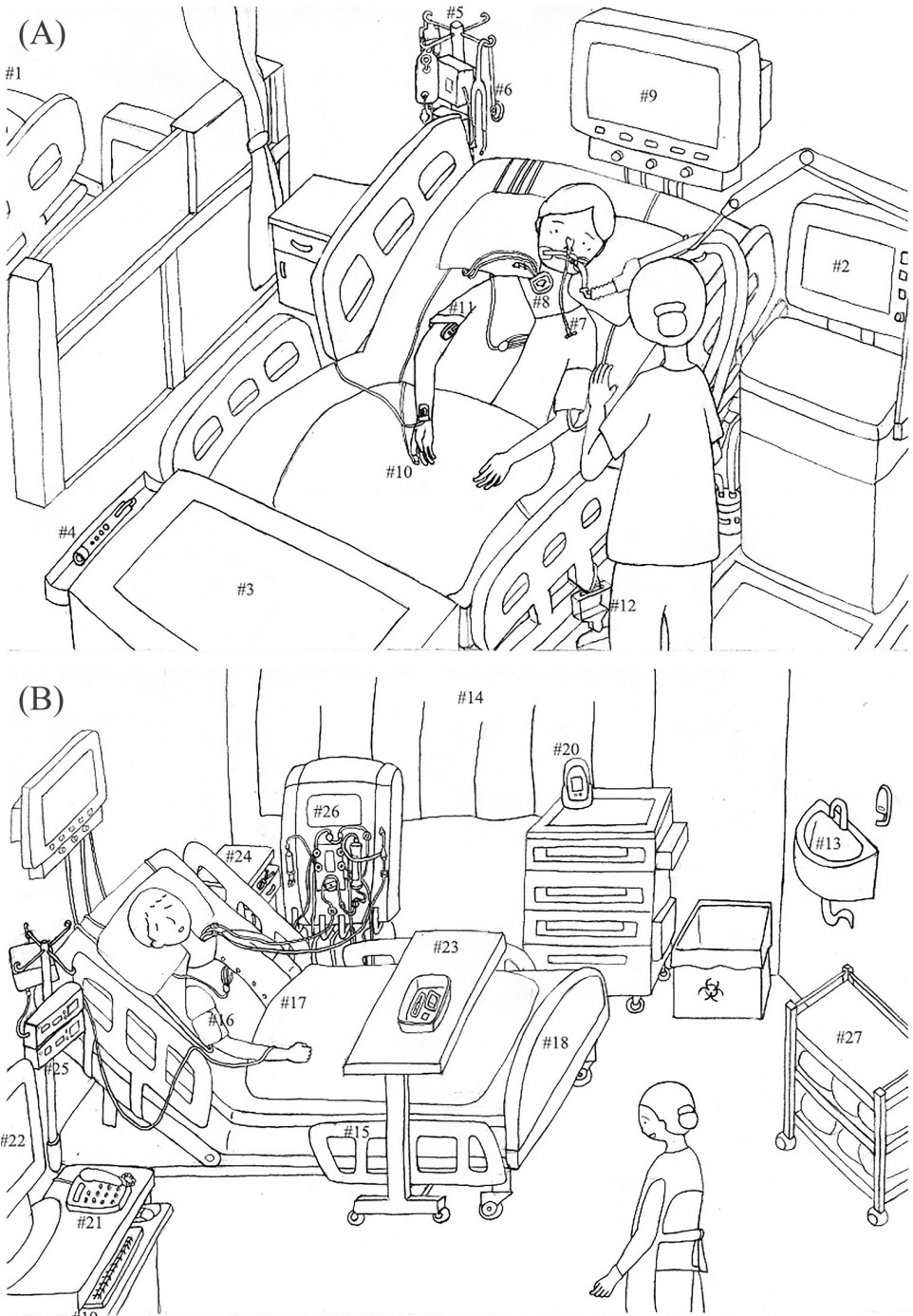

**Fig 1. Illustrations of the three nursing care scenarios.** (A) shows the locations of items in Scenario 1, which depicts open space. (B) shows the locations of items in Scenario 2 and 3, which depicts a private room.

For data analysis, the two IPC practitioners determined the correct answers for allocating the 27 items to their respective zones and subsequently validated them with a professor who has IPC expertise. For each item, the similarity among participants' responses was calculated by measuring the proportion of each participant's responses that matched the majority

choices, and the accuracy was determined by calculating the proportion of correct responses that concurred with the consensus reached by the IPC practitioners and professor. Similarity and accuracy were calculated for each participant to investigate potential variations based on their general characteristics. In addition to descriptive statistical analyses, the Wilcoxon rank-sum test or Kruskal-Wallis test for discrete variables and Spearman's rank correlation analysis for continuous variables were conducted using R version 4.0.2 (The R Foundation for Statistical Computing, Vienna, Austria). To delve into participants' opinions, thematic analysis was undertaken to identify recurring themes and patterns within their free-text responses.

## Results

### Participants' general characteristics

Of the 225 questionnaires distributed, 104 nurses completed the survey, resulting in a response rate of 46.2%. The participants' average age was 29.0 years, with an average work experience of 5.9 years (Table 1). Most participants (88.5%) were aware of the patient zone concept, mainly from hospitals (78.8%).

### Zone allocation of the items

Regarding categorising 27 items into zones, participants allocated 20 (74.1%) to the patient zone and 7 to the healthcare zone (Table 2). Most participants' responses regarding the curtain and glucometer diverged from the choices made by IPC practitioners and professors, indicating discrepancies in perspectives.

### Similarity and accuracy in items' zone allocation

When allocating zones for 27 items, the overall means for similarity and accuracy were 84.7% and 82.7%, respectively (Table 3). Among these, 12 items, which were either attached to the patient or were located nearby, showed more than 90% similarity and accuracy. For example, the pulse oximeter achieved 100% similarity and accuracy in allocation to the patient zone. However, among the seven items with similarity and accuracy lower than 74%, three (curtain, another patient, and sinks), which were located relatively distant from the patient, showed the lowest accuracy, ranging from 46.2% to 66.3%. Additionally, three other items (glucometers, flashlights, and trolleys [carts with wheels]) that were movable within the patient's proximity ranged from 70.2% to 73.1%. Notably, a glucometer located on a patient cart, which was positioned away from the patient within a private room, showed 73.1% similarity and 21.2% accuracy, while a patient cart located immediately next to the foot of the bed in the open area of the ICU showed 53.8% similarity and accuracy.

None of the participants achieved a 100% accuracy. However, participants with a master's degree or higher showed significantly higher similarity (*p* = .044) or accuracy (*p* = .033). Furthermore, participants with prior knowledge of the patient zone concept demonstrated significantly higher accuracy (*p* = .009; Table 4).

### Nurses' perceptions of items' zone allocation

Forty-four participants provided insights into the factors influencing the categorisation of items into different zones depending on various scenarios, such as whether the item was located within a private room or in the open ICU space, designated for exclusive use by a single patient or shared among multiple patients, and came into direct contact with the patient. In particular, portable items can be categorised differently because they often straddle both zones contingent on their current use or storage state. Furthermore, some participants indicated that

Table 1. Characteristics of 104 survey participants.

| Characteristics | Categories | No. (%) |
|---|---|---|
| **Age** | < 25 | 19 (18.3%) |
| | 25 ≤ ~ < 30 | 47 (45.2%) |
| | 30 ≤ ~ < 35 | 23 (22.1%) |
| | ≥ 35 | 15 (14.4%) |
| **Gender** | Male | 5 (4.8%) |
| | Female | 99 (95.2%) |
| **Total clinical experience (years)** | < 1 | 6 (5.8%) |
| | 1 ≤ ~ < 3 | 35 (33.7%) |
| | 3 ≤ ~ < 5 | 20 (19.2%) |
| | 5 ≤ ~ < 10 | 23 (22.1%) |
| | ≥ 10 | 20 (19.2%) |
| **ICU experience (years)** | < 1 | 6 (5.8%) |
| | 1 ≤ ~ < 3 | 38 (36.5%) |
| | 3 ≤ ~ < 5 | 20 (19.2%) |
| | 5 ≤ ~ < 10 | 21 (20.2%) |
| | ≥ 10 | 19 (18.3%) |
| **Department** | MICU1 | 18 (17.3%) |
| | SICU1 | 11 (10.6%) |
| | NRICU | 23 (22.1%) |
| | NSICU | 8 (7.7%) |
| | PICU1 | 15 (14.4%) |
| | PICU2 | 29 (27.9%) |
| **Level of Education** | ≤ Bachelor | 95 (91.3%) |
| | ≥ Master | 9 (8.7%) |
| **Prior knowledge of patient zone concept** | No | 11 (10.6%) |
| | Yes | 92 (88.5%) |
| | No response | 1 (1.0%) |
| **Learning route for the patient zone concept** | School only | 10 (9.6%) |
| | Hospital only | 66 (63.5%) |
| | School & Hospital | 15 (14.4%) |
| | Hospital & Others | 1 (1.0%) |
| | No response | 1 (1.0%) |
| | Not applicable | 11 (10.6%) |

ICU, intensive care unit; SD, standard deviation; MICU, medical intensive care unit; SICU, surgical intensive care unit; NRICU, neurological intensive care unit; NSICU, neurosurgical intensive care unit; PICU, paediatric intensive care unit

items such as a patient cart could potentially be considered the healthcare zone if they functioned as a station for medication preparation.

Of the 84 participants who responded to the criteria for categorising item zones, a common perspective emerged. Items in direct contact with patients were predominantly assigned to the patient zone. Additionally, the participants considered the item's proximity to the patient, potential for colonisation with patient flora, and need for disinfection, hand hygiene, or the use of personal protective equipment (PPE) when handling the item.

The challenges associated with item zone categorisation during routine tasks among the 48 respondents arose from a range of factors: confined spaces between the items, ambiguous

**Table 2. Allocation results for the 27 items by the participants.**

| Scenario numbers (patient room type) | Items | Patient Zone N (%) | Healthcare Zone N (%) | Difficult to decide N (%) |
|---|---|---|---|---|
| Scenario 1 (open ICU area) | #1 Another patient | 33 (31.7) | 60 (57.7) | 11 (10.6) |
| | #2 Ventilator monitor | 87 (83.7) | 17 (16.3) | 0 |
| | #3 Patient cart | 56 (53.8) | 48 (46.2) | 0 |
| | #4 Flashlight | 75 (72.1) | 28 (26.9) | 1 (1) |
| | #5 Pole | 93 (89.4) | 11 (10.6) | 0 |
| | #6 Stethoscope | 94 (90.4) | 10 (9.6) | 0 |
| | #7 Levin tube | 103 (99) | 1 (1) | 0 |
| | #8 Central line | 103 (99) | 1 (1) | 0 |
| | #9 Patient monitor | 89 (85.6) | 15 (14.4) | 0 |
| | #10 Pulse oximeter | 104 (100) | 0 | 0 |
| | #11 Thermometer | 99 (95.2) | 5 (4.8) | 0 |
| | #12 Urine bag | 103 (99) | 1 (1) | 0 |
| Scenario 2 (single-patient room) | #13 Sink | 23 (22.1) | 69 (66.3) | 12 (11.5) |
| | #14 Curtain | 50 (48.1)[a] | 48 (46.2) | 6 (5.8) |
| | #15 Bed rail | 101 (97.1) | 3 (2.9) | 0 |
| | #16 Blood pressure cuff | 102 (98.1) | 2 (1.9) | 0 |
| | #17 Linen | 101 (97.1) | 3 (2.9) | 0 |
| | #18 Patient bed | 103 (99) | 1 (1) | 0 |
| | #19 Keyboard | 9 (8.7) | 92 (88.5) | 3 (2.9) |
| Scenario 3 (single-patient room) | #20 Glucometer | 22 (21.2) | 76 (73.1)[a] | 6 (5.8) |
| | #21 Telephone fixed beside computer | 6 (5.8) | 94 (90.4) | 4 (3.8) |
| | #22 Computer | 8 (7.7) | 93 (89.4) | 3 (2.9) |
| | #23 Bedside table | 95 (91.3) | 9 (8.7) | 0 |
| | #24 Tourniquet | 87 (83.7) | 17 (16.3) | 0 |
| | #25 Infusion pump | 87 (83.7) | 16 (15.4) | 1 (1) |
| | #26 CRRT | 90 (86.5) | 13 (12.5) | 1 (1) |
| | #27 Trolley | 27 (26) | 73 (70.2) | 4 (3.8) |

[a]Majority answers are incorrect; ICU, intensive care unit; CRRT, continuous renal replacement therapy

boundaries, frequent relocation of portable items between usage and storage, ambiguous moments for hand hygiene, disinfection, or PPE use, and inconsistent standards across the hospital. While some individuals struggled with distinguishing zones within private rooms, others mentioned that the distinction was more difficult in open shared spaces. Moreover, items not directly in contact with patients but frequently handled by HCP presented their set of challenges.

To address these challenges, participants' suggestions from 31 respondents included the formulation of clear criteria supported by evidence-based guidelines and the provision of education for HCP. Practical solutions involve implementing visible spatial divisions using stickers, labels, and delineating lines. Furthermore, their suggested solutions include allocating dedicated equipment and expanding space.

## Discussion

This study demonstrated that ICU nurses had a limited perception of distinguishing between the patient and healthcare zones. This study also found that the participants considered a range of factors beyond mere proximity to the patient, making the determination of the patient

**Table 3. Similarity and accuracy of 27 items' zone allocation.**

| Items | Similarity (%) | Accuracy (%) |
|---|---|---|
| Pulse oximeter | 100.0 | 100.0 |
| Levin tube | 99.0 | 99.0 |
| Central line | 99.0 | 99.0 |
| Urine bag | 99.0 | 99.0 |
| Patient bed | 99.0 | 99.0 |
| Blood pressure cuff | 98.1 | 98.1 |
| Bed rail | 97.1 | 97.1 |
| Linen | 97.1 | 97.1 |
| Thermometer | 95.2 | 95.2 |
| Bedside table | 91.3 | 91.3 |
| Stethoscope | 90.4 | 90.4 |
| Telephone fixed beside computer | 90.4 | 90.4 |
| Pole | 89.4 | 89.4 |
| Computer | 89.4 | 89.4 |
| Keyboard | 88.5 | 88.5 |
| CRRT | 86.5 | 86.5 |
| Patient monitor | 85.6 | 85.6 |
| Ventilator monitor | 83.7 | 83.7 |
| Tourniquet | 83.7 | 83.7 |
| Infusion pump | 83.7 | 83.7 |
| Glucometer | 73.1 | 21.2 |
| Flashlight | 72.1 | 72.1 |
| Trolley | 70.2 | 70.2 |
| Sink | 66.3 | 66.3 |
| Another patient | 57.7 | 57.7 |
| Patient cart | 53.8 | 53.8 |
| Curtain | 48.1 | 46.2 |

CRRT, continuous renal replacement therapy

zone more intricate than its definition. ICU nurses may encounter challenges in distinguishing between zones based on criteria such as patient contact, room type, distance, cleanliness, and item portability.

The top eight items, all categorised within the patient zone, which maintained constant contact with patients and rarely left the patient's bed area in ICUs, showed over 97% similarity and accuracy. Given that patient contact was a commonly mentioned criterion among participants, classifying these items is relatively straightforward. Among these items, the central line, urine bag, bed rail, and linen were also categorised into the patient zone with high (100%) similarity in a previous study [24]. However, the blood pressure cuff showed only 70% similarity and 30% accuracy, as IPC experts allocated it to the healthcare zone in a previous study [24]. This might be due to more occasions for sharing a blood pressure cuff in general medical wards in that previous study compared designated single patient use of a blood pressure cuff in our ICUs, emphasising the importance of considering the context of item use across different healthcare settings when applying zone concepts.

Notably, the curtain had the lowest similarity (48.1%) and accuracy (46.2%), reflecting significant variation in our participants' perceptions. Some participants argued that it should belong to the patient zone if in an open area but to the healthcare zone if in a private room,

**Table 4. Similarity and accuracy according to the characteristics of the participants.**

| Characteristic | Similarity | | Accuracy | |
|---|---|---|---|---|
| | Mean ± SD | $W$ or $\chi^2$ or $r$ ($p$) | Mean ± SD | $W$ or $\chi^2$ or $r$ ($p$) |
| **Age** | | | | |
| $<$ **25** | 83.64 ± 9.50 | $\chi^2 = 2.62\ (0.455)$ | 80.92 ± 9.41 | $\chi^2 = 5.77\ (0.123)$ |
| **25 $\leq$ ~ $<$ 30** | 83.70 ± 11.60 | | 81.42 ± 11.72 | |
| **30 $\leq$ ~ $<$ 35** | 85.36 ± 12.86 | | 85.20 ± 12.98 | |
| $\geq$ **35** | 88.41 ± 5.39 | | 85.45 ± 5.67 | |
| **Gender** | | | | |
| **Male** | 89.64 ± 12.38 | $W = 168\ (0.223)$ | 87.42 ± 10.33 | $W = 162\ (0.191)$ |
| **Female** | 84.49 ± 10.78 | | 82.51 ± 11.03 | |
| **Total clinical experience (years)** | | | | |
| $<$ **1** | 80.88 ± 13.32 | $\chi^2 = 1.81\ (0.772)$ | 79.03 ± 11.14 | $\chi^2 = 2.78\ (0.595)$ |
| **1 $\leq$ ~ $<$ 3** | 84.46 ± 10.62 | | 81.39 ± 11.62 | |
| **3 $\leq$ ~ $<$ 5** | 83.53 ± 11.48 | | 82.98 ± 11.29 | |
| **5 $\leq$ ~ $<$ 10** | 84.71 ± 13.23 | | 83.59 ± 13.11 | |
| $\geq$ **10** | 87.61 ± 6.38 | | 85.02 ± 6.41 | |
| **ICU experience (years)** | | | | |
| $<$ **1** | 80.88 ± 13.32 | $\chi^2 = 2.50\ (0.644)$ | 79.03 ± 11.14 | $\chi^2 = 1.81\ (0.771)$ |
| **1 $\leq$ ~ $<$ 3** | 84.71 ± 10.30 | | 82.18 ± 11.51 | |
| **3 $\leq$ ~ $<$ 5** | 83.16 ± 11.29 | | 82.43 ± 10.86 | |
| **5 $\leq$ ~ $<$ 10** | 84.67 ± 13.83 | | 83.08 ± 13.64 | |
| $\geq$ **10** | 87.73 ± 6.53 | | 85.01 ± 6.58 | |
| **Department** | | | | |
| **MICU1** | 86.43 ± 4.40 | $\chi^2 = 8.79\ (0.117)$ | 85.61 ± 5.37 | $\chi^2 = 4.58\ (0.470)$ |
| **SICU1** | 88.23 ± 6.79 | | 85.20 ± 8.91 | |
| **NRICU** | 85.36 ± 13.65 | | 81.01 ± 13.55 | |
| **NSICU** | 90.29 ± 4.82 | | 87.98 ± 3.83 | |
| **PICU1** | 79.77 ± 15.71 | | 80.26 ± 16.49 | |
| **PICU2** | 82.90 ± 9.86 | | 81.24 ± 9.58 | |
| **Level of Education** | | | | |
| $\leq$ **Bachelor** | 84.15 ± 11.14 | $W = 255.5\ (0.044)$ | 82.12 ± 11.23 | $W = 245\ (0.033)$ |
| $\geq$ **Master** | 90.96 ± 3.26 | | 89.31 ± 4.70 | |
| **Prior knowledge of patient zone concept** | | | | |
| **No** | 78.46 ± 13.45 | $W = 330.5\ (0.058)$ | 74.09 ± 14.26 | $W = 265.5\ (0.009)$ |
| **Yes** | 85.60 ± 10.33 | | 83.87 ± 10.17 | |
| **Learning route for patient zone concept** | | | | |
| **School only** | 86.31 ± 8.55 | $\chi^2 = 0.41\ (0.938)$ | 84.09 ± 9.24 | $\chi^2 = 2.43\ (0.487)$ |
| **Hospital only** | 85.76 ± 10.75 | | 84.41 ± 10.60 | |
| **School & Hospital** | 84.21 ± 10.32 | | 81.50 ± 9.38 | |
| **Hospital & Others** | 88.90 | | 81.50 | |

ICU, intensive care unit; SD, standard deviation; MICU, medical intensive care unit; SICU, surgical intensive care unit; NRICU, neurological intensive care unit; NSICU, neurosurgical intensive care unit; PICU, paediatric intensive care unit

whereas others deemed everything within a private room to be the patient zone. However, more than 88% of the participants classified the telephone, computer, and keyboard within a private room as belonging to the healthcare zone, suggesting that most participants did not view everything within a private room as part of the patient zone. Meanwhile, some

participants found it more challenging to categorise items in open areas, emphasising the complexity that arises when physical boundaries are absent.

Participants considered the distance from patients when distinguishing between zones, which aligns with the definition of the patient zone as the immediate surroundings of the patient [17]. Consequently, participants also considered the positions of adjacent items, revealing the inherent complexity of categorising zones within the ICU. These participants' perceptions underscore how the placement of nearby items can affect zone allocation, highlighting the intricate interplay between items in patient and healthcare zones. This complexity suggests that patient proximity alone may be insufficient to distinguish between zones clearly. This may explain why some participants encountered difficulties when patient and healthcare zone items coexisted within the confined ICU spaces. Therefore, detailed criteria with explanations clarifying zone definitions must be developed to address the challenges of HCP.

In our study, items potentially contaminated by patient flora were typically categorised as the patient zone, whereas other items kept clean or sterile were generally considered the healthcare zone. This perspective aligns with the notion that the patient zone is susceptible to colonisation by flora, as supported by Sax et al. [17]. However, criticism exists regarding the idea of treating the area outside the patient zone as a single homogeneous zone because this oversimplifies the complexity of the healthcare environment [19]. In this study, we found a conflict between judgements based on the distance and cleanliness. For instance, concerning the patient cart, only a 7.6% difference was observed in the classification into the two zones, making it the second lowest similarity score after the curtain. Despite its proximity to the patient, some participants suggested that if the patient cart was used for medication preparation which required cleanliness, it should be categorised as a healthcare zone. Participants often categorised zones based on whether an item could be touched without necessitating hand hygiene after patient contact or wearing a gown after contact with an isolated patient. Disinfection requirements for an item were also identified as criteria in a previous study [24]. Nevertheless, the concept of the five hand hygiene moments primarily guides the timing of item disinfection rather than the inverse inference that the need for disinfection determines the item's zone [24]. Therefore, if cleanliness is not a suitable consideration, HCP education must address and correct misconceptions.

The difficulty in categorising zones ultimately leads to challenges in determining appropriate moments for hand hygiene [22], which can lead to patient harm [25]. The discrepancies observed between participants and IPC practitioners may result in the insufficient separation of microorganisms in clinical practice [24]. When reviewing items posing challenges for hand hygiene or item disinfection owing to zone differentiation, many were portable. Glucometer was the prime example of such an item and was misclassified as the healthcare zone by most participants (73.1%). In a previous study, mobile items posed the most significant allocation challenge because of their mobility across zones [24]. When determining zones for portable items is challenging, the risk of HAIs transmission increases. In a study monitoring 27 HCP over 39 hours to analyse behaviours with infection transmission risks, the third most common error out of 8 was the misplacement of items [28]. This resulted in contaminated items sometimes getting mixed with clean items and patient room items ending up in sterile areas [28]. In another study, low disinfection compliance was observed for items such as medication scanners that frequently moved in and out of the patient rooms [29]. As our two participants suggested for pupilometers, storing dedicated patient items within their designated patient zone could effectively reduce contamination risk.

To tackle the challenges associated with zone classification, it is imperative to establish well-defined criteria for patient and healthcare zones, consider space redesigns, and implement educational programs focusing on zone differentiation. In line with the study by Smith et al., where hand contact events were observed and the patient zone was redefined in an operating

room, IPC practitioners and frontline nurses should collaborate and reach a consensus on the definition of patient and healthcare zones within their respective departments [23]. Participants' suggestions to mark zones with stickers, tape, or lines on the floor align with recommendations from previous studies to improve participants' comprehension and mitigate contamination risk [25, 30]. Zone demarcation clarifies which item belongs to each zone and serves as a direct visual cue for HCP to perform hand hygiene [31]. As participants with higher education levels or prior knowledge displayed better accuracy, future research could explore the potential effectiveness of training programs. Such programs might enhance HCP's understanding of patient and healthcare zones, consequently improving their ability to determine the timing of hand hygiene and disinfection.

Our study has several limitations. Our study's response rate was relatively low (46.2%), likely due to nurses' fatigue caused by the exhaustive additional work required for the hospital's accreditation preparation during the data collection period, leaving little time or energy to complete our extensive scenario-based survey. Despite these barriers, nurses who volunteered may have possessed more interest in the patient zone concept, which could have influenced the study outcomes. As 88.5% of the participants had prior knowledge about the zones, nurses unfamiliar with the two zone concepts might have avoided participating in this study. While the nursing care scenarios were refined through expert validation, predicting item difficulty was inherently limited. Furthermore, the uneven distribution of items between zones in the scenarios, with only seven in the healthcare zone, might have introduced biased results. Future studies should include a broader spectrum of items and verified scenarios to obtain more precise and comprehensive results.

Given that this study was conducted in six ICUs at a single hospital in Korea, replicating it across diverse healthcare settings and among various HCP would provide more comprehensive insights, particularly for infection control practices. The use of detailed scenarios and illustrations in this study allowed nurses from different settings to gain a consistent understanding of the situations, enhancing the potential to generalise the findings to other hospital contexts or regions. Since item placement can significantly influence zone classification, hospitals should assess nurses' perceptions within the context of their specific environment and item arrangements. This approach will enable the development of optimised environmental management recommendations tailored to each hospital's unique setup, ultimately contributing to more effective infection control strategies.

## Conclusions

In our study, ICU nurses showed higher accuracy (82.7%) and similarity (84.7%) in distinguishing between patient and healthcare zones compared to a previous study conducted in a general medical ward, which reported 67.7% accuracy and 76.6% similarity [24]. Unlike the previous study using item cards, we adopted three nursing care scenarios with detailed illustrations showing exact item locations to examine ICU nurses' perceptions of the two zones, which is a notable strength of our approach. Consequently, our study revealed that participants could exhibit variations in their zones, reconfirming that the patient zone is not a geographically static area but rather varies depending on the setting and type of care delivered [32]. This observation highlights the need for further research to establish greater clarity and consensus regarding patient and healthcare zones to enhance infection control practices among HCP.

## Supporting information

**S1 File. The English version of the questionnaire.**
(DOCX)

## Acknowledgments

We would like to thank the healthcare personnel for participating in this study; nursing department officials and unit-managers from the participating hospital for their support in recruiting participants; and EunJi Kim, Songhee Namgung, Areum Yoo, and Sunjung Kim for evaluating the questionnaire and giving useful advice.

## Author Contributions

**Conceptualization:** Hayoung Chang, JaHyun Kang.

**Data curation:** Hayoung Chang.

**Formal analysis:** Hayoung Chang.

**Investigation:** Hayoung Chang.

**Methodology:** Hayoung Chang, JaHyun Kang.

**Project administration:** Hayoung Chang.

**Supervision:** JaHyun Kang.

**Validation:** JaHyun Kang.

**Visualization:** Hayoung Chang.

**Writing – original draft:** Hayoung Chang.

**Writing – review & editing:** JaHyun Kang.

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
