## [Decision Letter · Decision Letter 0]

17 Jul 2024

PONE-D-24-25721Perceptions of distinctions between patient and healthcare zones among intensive care unit nurses at a Korean tertiary hospital: a cross-sectional studyPLOS ONE

Dear Dr. Kang,

Thank you for submitting your manuscript to PLOS ONE. After careful consideration, we feel that it has merit but does not fully meet PLOS ONE’s publication criteria as it currently stands. Therefore, we invite you to submit a revised version of the manuscript that addresses the points raised during the review process.

The reviewers have provided constructive feedback to help enhance the clarity and rigor of your manuscript. We kindly ask that you carefully consider and address the following suggestions:

**Reviewer 1:**

**Abstract:** Add specific results for the similarity and accuracy scores.**Introduction:** Add aims and objectives in the introduction section.**Methods:** Include criteria used for calculating similarity and accuracy scores.   **Methods Optional**: Elaborate on the challenges faced during data collection and explain the process of assessing content validity during item selection. **Results Optional:** Use structured headings and subheadings to improve the clarity of the presentation.**Discussion:** Discuss whether you consider your results generalizable and why (what elements make them generalizable in your view and what elements limit their generalizability). Discuss also the broader implications for infection control practices.

**Reviewer 2:**

**Abstract:**
Provide a short but comprehensive background, offering a clear and concise overview of the research.Include details on the methodology, specifically the approach used and the number of participants recruited.Ensure statistical results (e.g., chi-square values, p-values) support the relationships between identified factors in the statement regarding factors considered by patients.
**Introduction (**point already highlighted by Reviewer 1**):**Clearly articulate the rationale and objectives for the study. Explicitly state the research problem or gap your study aims to address in the last paragraph of the introduction.
**Methodology:**
Clarify more how you arrive at a response rate of 46.2%. Did you include the total number of subjects eligible to participate in the denominator? Clarify the steps to arrive at 104 subjects.[Optional] Consider whether the description in Figure 1 should be left in the body of the article or it is better to include it as supplementary material.

Please revise your manuscript according to the reviewers’ comments and resubmit it for further consideration. We look forward to receiving your revised manuscript and are confident that these revisions will significantly strengthen your work.

Thank you once again for your valuable contribution. 

We look forward to receiving your revised manuscript.

Kind regards,

Lorenzo Righi

Academic Editor

PLOS ONE

Journal Requirements:

Reviewers' comments:

Reviewer's Responses to Questions

**Comments to the Author**

1. Is the manuscript technically sound, and do the data support the conclusions?

Reviewer #1: Yes

Reviewer #2: Yes

2. Has the statistical analysis been performed appropriately and rigorously? 

Reviewer #1: Yes

Reviewer #2: Yes

3. Have the authors made all data underlying the findings in their manuscript fully available?

Reviewer #1: Yes

Reviewer #2: Yes

4. Is the manuscript presented in an intelligible fashion and written in standard English?

Reviewer #1: Yes

Reviewer #2: Yes

5. Review Comments to the Author

Reviewer #1: Thanks for the opportunity to review this manuscript. My suggestions are as follows:

In the abstract, consider adding the specific results for the similarity and accuracy scores.

Lines 73 to 79 listing aims and objectives could be included in the end of the introduction section.

In the methods section, explain the challenges faced during data collection. This will strengthen the study's credibility. You could add the process of assessing content validity during item selection and the criteria used for calculating similarity and accuracy scores.

In the results section, you could enhance the clarity of the presentations using structured headings and subheadings.

In the discussion section, consider adding the following points.

The findings from this study are generalizable to other hospital contexts or regions. Also, it has broader implications for infection control practices beyond the study's specific hospital setting.

Reviewer #2: The manuscript requires minor corrections on the following section: abstract, introduction and methodology. as follows in the Abstract section: The background part should be comprehensive, highlighting a clear and concise overview of the research. The methodology part should include the approach used, number of participants recruited. I also noticed on factors considered by patients in this statement, ‘Participants considered factors beyond proximity to the patient, including patient contact, room type, distance, cleanliness, and item portability, in distinguishing between the two zones’ there are no statistical results (e.g., chi-square values, p-values) that support the relationships between the identified factors.

Additionally, the introduction section does not provide a clear rationale or objectives for the current study. While the authors have presented a comprehensive review of the existing literature, the introduction lacks a clear statement of the research problem or gap that this study aims to address. It is crucial to explicitly articulate the research objectives at the last paragraph of the introduction, as this sets the foundation for the entire study and helps the reader understand the significance and contributions of the work.

Lastly, the methodology section is generally well-described, with details provided on the study design, participant recruitment, and data collection procedures. However, I noticed that the manuscript does not include information on the number of participants recruited for the study. This is an important detail that should be included, as it allows the reader to assess the statistical power and generalizability of the findings. Also on page 6 where there is a description of an illustration, that part should be on the supplemental material, not the methodology part.

6. PLOS authors have the option to publish the peer review history of their article (what does this mean?). If published, this will include your full peer review and any attached files.

Reviewer #1: **Yes: **Manali Ulhas Desai

Reviewer #2: **Yes: **Tienyi Mnyoro Daniel

---

## [Author Response · Author response to Decision Letter 0]

14 Sep 2024

Dear Editor Righi:

Thank you for the reviewer comments on our manuscript entitled “Perceptions of distinctions between patient and healthcare zones among intensive care unit nurses at a Korean tertiary hospital: a cross-sectional study” (Ref: PONE-D-24-25721). We have revised the manuscript accordingly and our responses are as follows:

Reviewer #1: Thanks for the opportunity to review this manuscript. My suggestions are as follows:

In the abstract, consider adding the specific results for the similarity and accuracy scores. `

[Editor - Abstract: Add specific results for the similarity and accuracy scores.]

Thank you for the reviewer’s good point and an opportunity to make our abstract better. We added the specific results for the similarity and accuracy scores in the abstract like the below: “The top 8 items, with over 97% similarity and accuracy, were all frequently in contact with ICU patients (e.g., pulse oximeter, Levin tube, central line, urine bag, and patient bed). The bottom 7 items, with less than 80%, included the glucometer, flashlight, trolley, and sink.” Please note that we cannot include all 8 and 7 items due to the limitation of word counts, 300 words.

Lines 73 to 79 listing aims and objectives could be included in the end of the introduction section.

[Editor - Introduction: Add aims and objectives in the introduction section.]

Thank you for the reviewer’s comment. We have moved the following aims and objectives, originally described in the methods section, to the end of introduction line 75 to 81: “Given the limited prior research on this topic and the importance of HCP’s zone perceptions for infection control, we aimed to examine how the zone concept is applied in daily care activities by ICU nurses and obtain insights for improving infection control practices like hand hygiene or item disinfection. The study objectives were to assess ICU nurses’ perceptions of the zone concept and evaluate their accuracy and similarity in identifying the zones by presenting scenarios and illustrations depicting various patient care situations with item use.”

In the methods section, explain the challenges faced during data collection. This will strengthen the study's credibility. You could add the process of assessing content validity during item selection and the criteria used for calculating similarity and accuracy scores.

[Editor - Methods: Include criteria used for calculating similarity and accuracy scores.]

[Editor - Methods Optional: Elaborate on the challenges faced during data collection and explain the process of assessing content validity during item selection.]

Thank you for the suggestion. First, please note that we have already included a description of assessing content validity as follows: “Content validity was assessed by five area experts, all of whom held a master’s degree or higher: one nursing professor with infection prevention and control (IPC) expertise, two IPC practitioners, and two ICU nurses with more than 5 years of experience. After eliminating 17 items with a content validity index of 0.75 or less and 3 items with similar characteristics, the final 27 items were included (Fig 1).” (Line 110-116). To clarify content validity index calculations, we added “Items were rated as follows: 4 points for highly relevant, 3 points for somewhat relevant, 2 points for minimally relevant, and 1 point for not relevant.” in the middle of the relevant paragraph (Line 112-114).

Second, please also note that we have already described the criteria used for calculating similarity and accuracy in method section as follows: “For data analysis, the two IPC practitioners determined the correct answers for allocating the 27 items to their respective zones and subsequently validated them with a professor who has IPC expertise. For each item, similarity among participants’ responses was calculated by measuring the proportion of each participant’s responses that matched the majority choices, and accuracy was determined by calculating the proportion of correct responses that concurred with the consensus reached by the IPC practitioners and professor.” (Line 133-138) Additionally, the majority answers used for calculating similarity scores can be found in Table 2, with any instances where the majority answer was incorrect noted in the table's notes. The correct answers needed to calculate accuracy scores can also be identified in Table 2. 

Third, in response to the reviewer’s comment, we have added the challenges faced during data collection to the discussion section instead of methods section for logical flow, as follows: “Our study’s response rate was relatively low (46.2%), likely due to nurses' fatigue caused by the exhaustive additional work required for the hospital's accreditation preparation during the data collection period, leaving little time or energy to complete our extensive scenario-based survey.”

In the results section, you could enhance the clarity of the presentations using structured headings and subheadings.

[Editor - Results Optional: Use structured headings and subheadings to improve the clarity of the presentation.]

Thank you for the suggestion. Accordingly, we have added four subheadings to the result section as follows:

Participants’ general characteristics

Zone allocation of the items

Similarity and accuracy in items’ zone allocation

Nurses’ perceptions of the items’ zone allocation

In the discussion section, consider adding the following points.

The findings from this study are generalizable to other hospital contexts or regions. Also, it has broader implications for infection control practices beyond the study's specific hospital setting.

[Editor - Discussion: Discuss whether you consider your results generalizable and why (what elements make them generalizable in your view and what elements limit their generalizability). Discuss also the broader implications for infection control practices.]

Thank you for the great point. We have changed and added more in the discussion section as follows:

Before Revision - As this study was conducted within six ICUs at a single hospital in Korea, replicating with diverse backgrounds and among various HCP would be beneficial.

After Revision - Given that this study was conducted in six ICUs at a single hospital in Korea, replicating it across diverse healthcare settings and among various HCP would provide more comprehensive insights, particularly for infection control practices. The use of detailed scenarios and illustrations in this study allowed nurses from different settings to gain a consistent understanding of the situations, enhancing the potential to generalise the findings to other hospital contexts or regions. Since item placement can significantly influence zone classification, hospitals should assess nurses’ perceptions within the context of their specific environment and item arrangements. This approach will enable the development of optimised environmental management recommendations tailored to each hospital’s unique setup, ultimately contributing to more effective infection control strategies.

Reviewer #2: The manuscript requires minor corrections on the following section: abstract, introduction and methodology. as follows in the Abstract section: The background part should be comprehensive, highlighting a clear and concise overview of the research. 

[Editor - Abstract: Provide a short but comprehensive background, offering a clear and concise overview of the research.]

Thank you for the reviewer’s valuable feedback and the opportunity to improve our abstract. We have revised the background section of the abstract as follows: 

Before Revision - Intensive care unit nurses must distinguish between patient and healthcare zones for effective infection control; however, there is limited research on this topic.

After Revision -Intensive care unit (ICU) patients face higher infection risks from invasive procedures, highlighting the critical role of ICU nurses in infection prevention. Clear differentiation between the patient zone and healthcare zone is essential for effective hand hygiene and disinfection, yet research on this topic is limited.

The methodology part should include the approach used, number of participants recruited.

[Editor – Abstract: Include details on the methodology, specifically the approach used and the number of participants recruited.]

Thank you for the opportunity to improve our methods section with your valuable suggestion. We have revised the methods section as follows: 

Before Revision - Written informed consent was collected from all participants. Through flyers posted at nursing stations, participants were recruited from ICUs, including medical, surgical, neurological, neurosurgical, and paediatric ICUs. Nurses with at least 2 months of independent work experience providing direct patient care within ICUs were eligible for this study. Nurse administrators who did not engage in direct patient care were also excluded.

After Revision - Nurses with at least 2 months of independent work experience providing direct patient care within ICUs were eligible for this study. Nurse administrators who did not engage in direct patient care were also excluded. Participants were recruited from various ICUs—medical, surgical, neurological, neurosurgical, and paediatric— through flyers posted at nursing stations. A minimum of 180 participants was needed to analyse significant differences based on general characteristics. To account for a 20% dropout rate, the target sample size was set at 225 participants. Accordingly, 225 printed consent forms and questionnaires were made available at the ICU nursing stations, allowing nurses to access and complete them voluntarily. Written informed consent was collected from all participants.

Accordingly, we have also revised the methods section in the abstract as follows: 

Before Revision - A descriptive survey was conducted at a 2,732-bed tertiary hospital in Korea from 28 July to 27 August 2022, with illustrations for three intensive care unit nursing care scenarios providing a clear depiction of 27 item locations. Nurses working in medical, surgical, neurological, neurosurgical, and paediatric ICUs were recruited. The similarity score was calculated as the proportion of each participant’s responses that matched most participants’ choices, and the accuracy score was calculated as the proportion of correct answers.

After Revision - A descriptive survey was conducted at a 2,732-bed tertiary hospital in Korea from 28 July to 27 August 2022. Participants were recruited from various ICUs through flyers. 225 questionnaires—with illustrations 27 item locations for three ICU scenarios—were made available at nursing stations for voluntary completion. Participants were asked to classify the items into the patient zone or the healthcare zone. Similarity scores were reflected participant agreement, while accuracy scores measured the proportion of correct answers.

I also noticed on factors considered by patients in this statement, ‘Participants considered factors beyond proximity to the patient, including patient contact, room type, distance, cleanliness, and item portability, in distinguishing between the two zones’ there are no statistical results (e.g., chi-square values, p-values) that support the relationships between the identified factors.

[Editor – Abstract: Ensure statistical results (e.g., chi-square values, p-values) support the relationships between identified factors in the statement regarding factors considered by patients.]

Please note that the factors considered by participants when identifying the zones were summarized from thematic analysis for participants’ free-text responses as described in the methods section as follows: “At the end of the survey, four additional open-ended questions were asked.” and “To delve into participants’ opinions, thematic analysis was undertaken to identify recurring themes and patterns within their free-text responses.” Thus, statistical analysis could not be applied. To avoid audience’s misunderstandings, we added the phrase, “From the free-text analysis” to the beginning of the sentence in the abstract.

Before Revision - Participants considered factors beyond proximity to the patient, including patient contact, room type, distance, cleanliness, and item portability, in distinguishing between the two zones.

After Revision - From the free-text analysis, participants considered factors beyond proximity to the patient, including patient contact, room type, distance, cleanliness, and item portability, in distinguishing between the two zones.

Additionally, the introduction section does not provide a clear rationale or objectives for the current study. While the authors have presented a comprehensive review of the existing literature, the introduction lacks a clear statement of the research problem or gap that this study aims to address. It is crucial to explicitly articulate the research objectives at the last paragraph of the introduction, as this sets the foundation for the entire study and helps the reader understand the significance and contributions of the work.

[Editor - Introduction (point already highlighted by Reviewer 1): Clearly articulate the rationale and objectives for the study. Explicitly state the research problem or gap your study aims to address in the last paragraph of the introduction.]

Thank you for the suggestion. Reviewer 1 also provided the same suggestion. As we responded to the reviewer 1’s comment, we have moved the aims and objectives, originally described in the methods section to the end of introduction (line 75 to 81): “Given the limited prior research on this topic and the importance of HCP’s zone perceptions for infection control, we aimed to examine how the zone concept is applied in daily care activities by ICU nurses and obtain insights for improving infection control practices like hand hygiene or item disinfection. The study objectives were to assess ICU nurses’ perceptions of the zone concept and evaluate their accuracy and similarity in identifying the zones by presenting scenarios and illustrations depicting various patient care situations with item use.”

Lastly, the methodology section is generally well-described, with details provided on the study design, participant recruitment, and data collection procedures. However, I noticed that the manuscript does not include information on the number of participants recruited for the study. This is an important detail that should be included, as it allows the reader to assess the statistical power and generalizability of the findings. 

[Editor - Methodology: Clarify more how you arrive at a response rate of 46.2%. Did you include the total number of subjects eligible to participate in the denominator? Clarify the steps to arrive at 104 subjects.]

Thank you for the suggestion. This point is similar to the 2nd comment from Reviewer 2, which we have already addressed above. To clarify the denominator, we have added the number of distributed questionnaires and explained how participants were recruited in the methods section as follows: “A minimum of 180 participants was needed to analyse significant differences based on general characteristics. To account for a 20% dropout rate, the target sample size was set at 225 participants. Accordingly, 225 printed consent forms and questionnaires were made available at the ICU nursing stations, allowing nurses to access and complete them voluntarily.”

To clarify the denominator, we have also revised the results sentence as follows:

Before Revision - One hundred and four nurses completed the survey, resulting in a response rate of 46.2%.

After Revision - Of the 225 questionnaires distributed, 104 nurses completed the survey, resulting in a response rate of 46.2%.

Also on page 6 where there is a description of an illustration, that part should be on the supplemental material, not the methodology part.

[Editor - [Optional] Consider whether the description in Figure 1 should be left in the body of the article or it is better to include it as supplementary material.]

Thank you for the suggestion to improve our manuscript. We have removed the description of each item in Figure 1 note from the text, as this 

---

## [Editor Report · Decision Letter 1]

18 Sep 2024

Perceptions of distinctions between patient and healthcare zones among intensive care unit nurses at a Korean tertiary hospital: a cross-sectional study

PONE-D-24-25721R1

Dear Dr. Kang,

We’re pleased to inform you that your manuscript has been judged scientifically suitable for publication and will be formally accepted for publication once it meets all outstanding technical requirements.

Kind regards,

Lorenzo Righi

Academic Editor

PLOS ONE
---

## [Editor Report · Acceptance letter]

23 Sep 2024

PONE-D-24-25721R1 

PLOS ONE

Dear Dr. Kang, 

I'm pleased to inform you that your manuscript has been deemed suitable for publication in PLOS ONE. Congratulations! Your manuscript is now being handed over to our production team.

Kind regards, 

on behalf of

Dr. Lorenzo Righi 

Academic Editor

PLOS ONE